# DEveloping Tests for Endometrial Cancer deTection (DETECT): protocol for a diagnostic accuracy study of urine and vaginal samples for the detection of endometrial cancer by cytology in women with postmenopausal bleeding

Eleanor R Jones,[1,2] Suzanne Carter,[1] Helena O'Flynn,[1] Kelechi Njoku,[1]
Chloe E Barr,[1,2] Nadira Narine,[3] David Shelton,[3] Durgesh Rana,[3]
Emma J Crosbie [1,2]

For numbered affiliations see end of article.

**Correspondence to**
Professor Emma J Crosbie;
emma.crosbie@manchester.ac.uk

## ABSTRACT

**Introduction** Postmenopausal bleeding (PMB), the red flag symptom for endometrial cancer, triggers urgent investigation by transvaginal ultrasound scan, hysteroscopy and/or endometrial biopsy. These investigations are costly, invasive and often painful or distressing for women. In a pilot study, we found that voided urine and non-invasive vaginal samples from women with endometrial cancer contain malignant cells that can be identified by cytology. The aim of the DEveloping Tests for Endometrial Cancer deTection (DETECT) Study is to determine the diagnostic test accuracy of urine and vaginal cytology for endometrial cancer detection in women with PMB.

**Methods and analysis** This is a multicentre diagnostic accuracy study of women referred to secondary care with PMB. Eligible women will be asked to provide a self-collected voided urine sample and a vaginal sample collected with a Delphi screener before routine clinical procedures. Pairs of specialist cytologists, blinded to participant cancer status, will assess and classify samples independently, with differences settled by consensus review or involving a third cytologist. Results will be compared with clinical outcomes from standard diagnostic tests. A sample size of 2000 women will have 80% power to establish a sensitivity of vaginal samples for endometrial cancer detection by cytology of ≥85%±7%, assuming 5% endometrial cancer prevalence. The primary objective is to determine the diagnostic accuracy of urogenital samples for endometrial cancer detection by cytology. Secondary objectives include the acceptability of urine and vaginal sampling to women.

**Ethics and dissemination** This study has been approved by the North West–Greater Manchester West Research Ethics Committee (16/NW/0660) and the Health Research Authority. Results will be disseminated through publication in peer-reviewed scientific journals, presentation at conferences and via charity websites.

**Trial registration number** ISRCTN58863784.

### Strengths and limitations of this study

► This is a prospective evaluation of a novel, non-invasive endometrial cancer detection tool that could transform diagnostic pathways for women with postmenopausal bleeding (PMB).

► Samples will be taken prior to routine clinical procedures to avoid inadvertent contamination of samples by iatrogenically dislodged endometrial cells.

► Cytologists are blinded to participant cancer status until they provide their consensus report.

► Passive follow-up of participants will ensure missed cancer diagnoses are minimised.

► Recruitment is limited to women with PMB, and results may not be applicable to premenopausal women or those with atypical presentations of endometrial cancer.

## INTRODUCTION

In the UK, over 9000 women are diagnosed with endometrial cancer every year.[1] The red flag symptom for endometrial cancer is postmenopausal bleeding (PMB).[2] A woman with PMB is referred by her general practitioner (GP) on the urgent 'suspected cancer' pathway to a rapid access gynaecology clinic, where she is offered a series of invasive, unpleasant and often painful tests to rule out endometrial cancer.[3] These include transvaginal sonography (TVS), outpatient hysteroscopy and endometrial biopsy.[4] Together, these tests cost the National Health Service (NHS) around £750/woman.[5] PMB is extremely common, and only 5%–10% of women with PMB are ultimately diagnosed with endometrial cancer.[6 7] Indeed, it has been estimated that 5% of all GP referrals to gynaecology, as

many as 150 000 women per year in the UK alone, relate to PMB.[5] A simple, non-invasive test deployed in primary care to target those at risk of endometrial cancer for invasive testing, while safely reassuring the vast majority of healthy women, could transform diagnostic pathways for endometrial cancer. In the UK, it would save thousands of women every year from the psychological and physical sequelae of invasive tests and create substantial cost savings for the NHS (potential saving in excess of £100 million/year).

The development of novel non-invasive detection tools was voted the most important research priority for detecting cancer early in the recently completed James Lind Alliance Priority Setting Partnership.[8] In endometrial cancer, the anatomical continuity between the uterus and the lower genital tract facilitates the collection of shed tumour cells using non-invasive sampling methodologies.[9–13] We found that shed tumour cells can be collected from the vagina by gentle lavage using the Delphi screener and from voided urine samples, which are inevitably contaminated by endometrial debris in women with uterine bleeding.[14] These cells can be distinguished from normal squamous, urothelial and glandular cells of the urogenital tract by cytology, although certain benign mimics (eg, polyps and atrophy) can cause difficulties with interpretation. In our pilot study of 113 women with unexplained PMB, urine and/or vaginal cytology showed 100% sensitivity and 100% negative predictive value (NPV) for endometrial cancer detection; it identified all seven cancers (four endometrial, one cervical, one ovarian and one bladder) for a 11% false positive rate. Furthermore, mean pain scores were significantly lower for vaginal sampling (1.61, SD 2.04) than for diagnostic hysteroscopy (4.28, SD 2.61, p<0.001) and endometrial biopsy (4.88, SD 3.49, p<0.001), respectively.[14] Thus, urogenital cytology has considerable potential as a well-tolerated 'rule out' test to enable quick reassurance for most women who present to primary care with PMB and urgent referral for those who test positive. To confirm its clinical utility for endometrial cancer detection, urine and vaginal cytology must now be tested in a large prospective study of women undergoing investigation for PMB. The aim of the DEveloping Tests for Endometrial Cancer deTection (DETECT) Study is to estimate the diagnostic accuracy of urine and vaginal samples for endometrial cancer detection by cytology in women with PMB.

## METHODS
This protocol is reported in accordance with Standards for Reporting of Diagnostic Accuracy Studies (STARD) guidelines.[15]

## Study design
DETECT is a prospective multicentre diagnostic accuracy study of urine and vaginal cytology for endometrial cancer detection in women with PMB (study schema, figure 1).

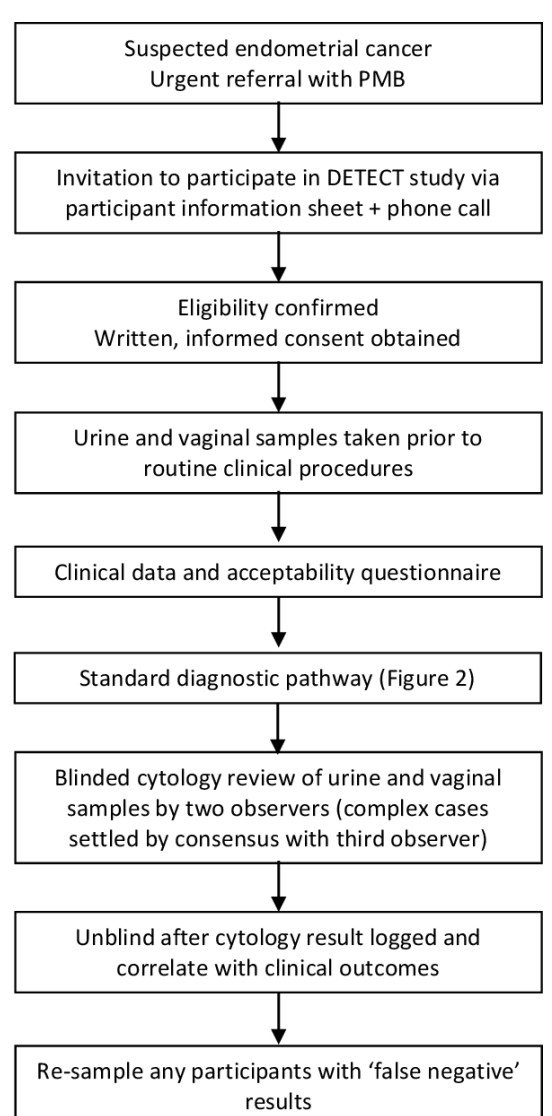

**Figure 1** Study schema illustrating the flow of participants through the study, interventions and evaluations. DETECT, DEveloping Tests for Endometrial Cancer deTection; PMB, postmenopausal bleeding.

## Participants and recruitment
Consecutive women will be recruited from gynaecology clinics at seven hospital sites across the North West England: St Mary's Hospital, Trafford General Hospital, Wythenshawe Hospital, Royal Oldham Hospital, North Manchester General Hospital, Fairfield General Hospital and Tameside Hospital.

## Inclusion criteria
1. Women who have been referred to secondary care for investigation of PMB.
2. Written, informed consent to participate.

## Exclusion criteria
1. Abnormal bleeding before the menopause.
2. Previous treatment for endometrial cancer.
3. Previous hysterectomy.
4. Mirena coil in situ or removed within the last 3 months.

5. Any other condition that would compromise participant safety or data integrity.

PMB will be defined as vaginal bleeding that occurs more than 12 months after menstruation has stopped due to menopause.

### Participant withdrawal

Participants may withdraw from the study at their own request or at the discretion of the investigator. Withdrawal from the study will not affect patient care.

### Sample size

With a sample size of ~2000 women and an endometrial cancer prevalence of 5%,[5] there will be approximately 1900 women without endometrial cancer and 100 women with endometrial cancer. The study will have 80% power to determine the sensitivity of the test at ≥85%±7% and the specificity of the test at ≥85%±2%. At 85% specificity, around 1630 women will have a negative test result, giving the test a NPV of 99.1% (98.5% and 99.4%). These estimates originate from the pilot study, where sensitivity and specificity were both >85%. The prevalence of endometrial cancer will determine the final sample size. If the prevalence of endometrial cancer is greater than 5%, fewer than 2000 women will be needed; if it is less than 5%, more women will be needed to determine the sensitivity of the test at ≥85%±7%.

### Study duration

In the pilot study, 90% of eligible women agreed to participate. Women will be recruited between September 2018 and September 2021 with clinical outcome data collected until March 2022. To achieve the recruitment target of ~2000 women, approximately 20 women will be recruited per week across the seven hospital sites. A temporary pause on recruitment to the study was initiated in March 2020 because of the COVID-19 pandemic. Recruitment recommenced in June 2020 but with COVID-19 restrictions in place, meaning that recruitment rate is slower than pre-pandemic (approximately 10 women per week across recruitment sites).

### Invitational procedure

Eligible women will be identified from referral letters and clinic lists by hospital clinical staff. A letter inviting participation and a detailed participant information sheet (PIS) will be sent by post to their home address. Potential participants will have the opportunity to discuss the study over the telephone with members of the study research team prior to their appointment. Due to the urgent nature of referrals, some women are offered short-notice clinical appointments by telephone. Where there is insufficient time for invitational material to be received by post, women will be informed about the study via telephone and/or invited to participate on arrival at the clinic. Women will be invited to read the PIS and ask questions about the study before providing written, informed consent to participate.

### Baseline clinical data

The following demographical and baseline clinical data will be obtained: age, ethnicity, socioeconomic status, education level, body mass index (BMI), smoking status, parity, age of menopause, use of contraceptives, hormone replacement therapy or tamoxifen, history of hypertension, hypercholesterolaemia, diabetes mellitus, polycystic ovary syndrome, thyroid disease, coagulopathy, Lynch syndrome, cervical screening history, endometrial hyperplasia and personal or family history of cancer. Women will be asked in detail about their help-seeking behaviour and the onset, duration and extent of their PMB.

### Index tests

The timing of urine and vaginal sampling is important to ensure validity of the results. Both research samples will be taken before any routine clinical procedures to avoid inadvertent contamination with iatrogenically dislodged endometrial cells (study schema, figure 1). Of the two research samples, urine will be collected first because vaginal sampling will remove naturally shed uterine debris from the lower genital tract and affect interpretation of the results.

### Urine samples

Using both written and verbal instructions, we will ask women to bring to clinic a first catch sample of their first urinary void of the day, collected in a sterile pot. A second voided urine sample will be collected by the participant on arrival at the clinic, before any other research or clinical procedures are carried out. This will ensure that every woman has at least one satisfactory urine sample available for analysis.

### Vaginal sample

The vaginal sample will be taken by the research practitioner using a Delphi screener (Rovers Medical Devices, Oss, the Netherlands)[16] according to the following protocol, with the participant in the supine position, knees bent and legs apart. The Delphi screener is inserted into the posterior fornix of the vagina and 3 mL saline expelled from its reservoir by depressing the plunger for 3 s. The sample is then collected by suction following release of the plunger while slowly rotating and withdrawing the device. A dry pot at the introitus collects any residual fluid. Additional samples are obtained by refilling the reservoir with saline and repeating the steps above until clear fluid is obtained (maximum of three times).

### Sample handling

Urine samples will be tested for haematuria by dipstick. Urine and vaginal samples will be fixed with equal volumes of BD CytoRich Red Preservative (Becton, Dickinson and Company, the USA) to preserve cellular integrity, prevent degradation and inhibit bacterial overgrowth. Samples will be sent to the Manchester Cytology Centre at the Manchester University NHS Foundation Trust (MFT). Samples will be anonymised and labelled with sample type (urine or vaginal fluid) and a unique study identifier

(study ID) to prevent accidental unblinding of the cytopathology team. With the exception of whether the participant is taking exogenous hormones, all clinical data will be withheld from the cytology team until consensus results are received. Second urine samples will be fixed with BD CytoRich Red Preservative and stored at 4°C until the first urine sample has undergone cytological assessment. This second sample will undergo cytological review if the first urine sample is inadequate for analysis or post hoc, if the first urine sample is false negative for endometrial cancer detection. Residual samples will be stored in the MFT Biobank for future biomarker discovery work. Samples will be either embedded in agar cell blocks to preserve cellular integrity for future immunocytochemistry or centrifuged to pellet the cellular material. The resulting pellet plus aliquots of the supernatant will be stored at −80°C.

## Cytological assessment

Samples will be centrifuged at 3000 revolutions per minute (RPM) for 5 min, supernatant decanted and the pellet resuspended in 6 mL BD CytoRich Red Preservative. After 1 hour, the fixed sample will be centrifuged at 1500 RPM for 10 min, supernatant decanted and the remaining pellet prepared into a liquid-based cytology Papanicolaou stained slide using the BD PrepStain (Becton Dickinson UK) according to the manufacturer's instructions. The stained slide will be dehydrated in two changes of industrial methylated spirits, cleared in two changes of xylene and cover-slipped. Two observers, a consultant cytopathologist and a consultant biomedical scientist, will review each slide independently and record their results. A second consultant cytopathologist will review any discrepant cases, which will be settled by consensus review at a multiheaded microscope. Final cytology results will be logged under the unique study ID on the study database. intraobserver and interobserver variability will be determined by blinded, independent review of a random selection of anonymised test positive, test negative, complex and discrepant cases at the end of the study and reported separately.

## Classification of cytology results

Cytology slides will be reported according to the classification system shown in table 1. For the primary analysis, atypical cells of uncertain significance (ACUS), suspicious, adenocarcinoma or malignant (other) cytology results will be considered positive. Glandular cells and no malignant cells seen will be considered negative. Unsatisfactory results will not be classified as either positive or negative, and the participant will be invited to provide a second sample for cytological analysis. Secondary analysis will include cancers of other pelvic sites (cervix, vagina, uterus, ovary, fallopian tube, bladder, kidney or bowel). Sensitivity analysis will consider the diagnostic performance of positive urogenital cytology that includes glandular results as potentially malignant findings.

## Clinical diagnostic pathway

Women will be investigated by TVS, outpatient hysteroscopy and/or endometrial biopsy according to local clinical diagnostic pathways for the investigation of PMB (figure 2).[3] All women will have their endometrium measured by TVS. Those with an endometrial thickness <4 mm will be considered at low risk of endometrial cancer and alternative diagnoses explored. Women with ≥4 mm endometrial thickness will undergo an endometrial biopsy using a pipelle endometrial sampler. Those with an irregular thickened endometrium, where focal pathology is visualised or suspected, will have an outpatient hysteroscopy and suspicious lesions biopsied under direct vision. Hysteroscopy will be performed under general anaesthesia where outpatient hysteroscopy fails, biopsies are inadequate for diagnostic purposes or uterine instrumentation is poorly tolerated. Endometrial polyps will be resected to allow full histological interpretation. Tissue samples will be formalin-fixed, paraffin-embedded, cut into 4 μm sections, stained with H&E and cover-slipped as per routine practice. At least one pathologist will review all biopsies; suspicious or abnormal biopsies will be reviewed by two specialist gynaecological pathologists at the cancer centre (St Mary's Hospital) as per routine clinical practice; difficult cases will be reviewed by additional specialist members of the gynaecological pathology team. Hysterectomy specimens will be fixed, cut, sectioned and

| Table 1 | Cytological classification | | |
|---|---|---|---|
| **Cytology result** | **Cytological findings** | **Primary analysis** | **Sensitivity analysis** |
| Unsatisfactory* | Sample obscured by debris, lymphocytes or bacteria | Indeterminate | Indeterminate |
| NMCS | No malignant cells seen | Negative | Negative |
| Glandular cells | Endometrial glandular cells seen | Negative | Positive |
| ACUS | Atypical cells of uncertain significance | Positive | Positive |
| Suspicious | Atypical glandular cells, suspicious for malignancy | Positive | Positive |
| Adenocarcinoma | Adenocarcinoma malignant cells seen | Positive | Positive |
| Malignant (others) | Malignant cells of non-endometrial origin | Positive | Positive |

*Urine according to the Paris criteria.

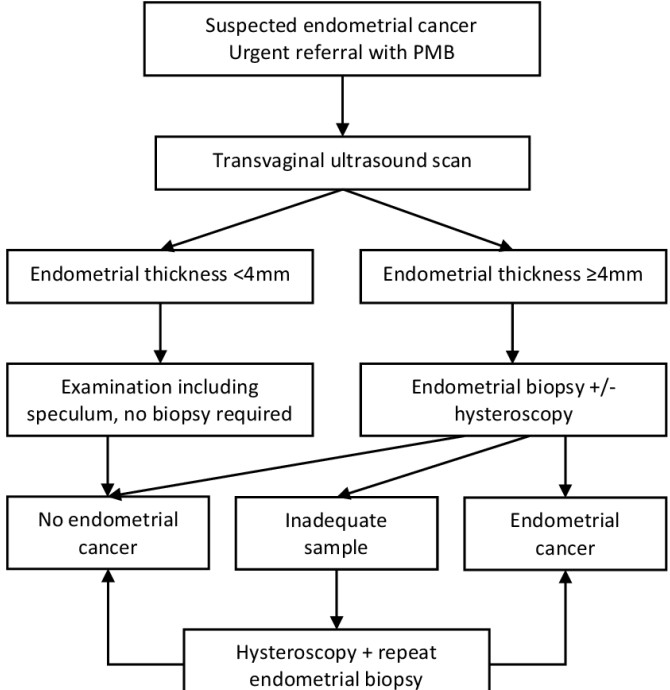

**Figure 2** Diagnostic pathway for women referred to secondary care for the urgent investigation of unexplained postmenopausal bleeding (PMB).

reviewed according to the International Federation of Gynecology and Obstetrics (FIGO) 2009 staging criteria (endometrial cancer) and the WHO classification system (atypical hyperplasia).[17][18]

## Reference standard

The reference standard is histology where endometrial tissue is collected for routine diagnostic and staging purposes. The hysterectomy specimen will be used in preference to the endometrial biopsy, where available within 3 months of endometrial biopsy and where there was no treatment with neoadjuvant chemotherapy or hormone therapy. Histology results will be classified as inadequate, benign, atypical hyperplasia and endometrial cancer (of any histological subtype or grade). For the primary analysis, a histological diagnosis of endometrial cancer will be considered a positive result. We will report overall results as well as a breakdown by histological subtype and stage of disease. Hyperplasia and benign endometrium will be considered a negative result. In secondary analyses, cancers of other pelvic sites (cervix, vagina, uterus, ovary, fallopian tube, bladder, kidney or bowel) will be considered a positive result. We will record cases of atypical hyperplasia and their cytological interpretation. In cases where an endometrial biopsy is not indicated (endometrial thickness <4 mm and/or normal hysteroscopy) or fails (inadequate sample), the reference standard will be discharge from diagnostic workup. Passive clinical follow-up of negative women will ensure missed diagnoses of endometrial cancer are minimised. This will involve monitoring for any subsequent rereferrals to our service. A diagnosis of endometrial cancer within 3 months of initial discharge from diagnostic workup will be considered a positive clinical result.

## Blinding

In most cases, research samples will be collected prior to routine diagnostic workup, and sample takers will be blinded to participant cancer outcomes. If the proportion of cases is less than 5% at the midpoint of the study, the cohort will be enriched with endometrial cancer diagnoses by recruiting higher-risk women (endometrial thickness >4 mm on TVS) and those with proven endometrial cancer prior to hysterectomy. Care will be taken to collect research samples at least 7 days after any diagnostic uterine instrumentation. The cytology team will always be blinded to the cancer status of participants when performing cytological review of urine and vaginal samples. They will have no access to clinical data or results of routine diagnostic tests. Individual participants will not routinely receive their cytology results.

## Outcome measures
### Primary outcome measure

*Sensitivity*—the proportion of women with endometrial cancer who test positive by cytology (true positive rate) and *specificity*—the proportion of women who do not have endometrial cancer who test negative by cytology (true negative rate). The accuracy of cytology (index test) will be defined by the results of standard endometrial cancer diagnostic tests (reference standard, figure 2).

## Secondary outcome measures

1. NPV—the proportion of women with a negative test who do not have endometrial cancer and *positive predictive value*—the proportion of women with a positive test who have endometrial cancer.
2. False positive/negative rates (and clinical scenarios associated with these).
3. Overall diagnostic accuracy of cytology for endometrial cancer detection.
4. Clinical performance of cytology for the detection of endometrial, cervical or bladder cancer.
5. Clinical performance of cytology for the detection of any pelvic cancer.
6. Test acceptability (short questionnaire to compare acceptability of urogenital sampling with standard diagnostic tests in a proportion of participants, eg, 5%–10%).

## Vaginal or urine cytology

1. Sensitivity and specificity of vaginal or urine cytology alone for endometrial cancer detection.
2. NPV, positive predictive value and false positive/negative rates of vaginal or urine cytology alone for endometrial cancer detection.
3. Overall diagnostic accuracy of vaginal or urine cytology alone for endometrial cancer detection.
4. Clinical performance of vaginal or urine cytology alone for the detection of endometrial, cervical or bladder cancer.

5. Clinical performance of vaginal or urine cytology alone for the detection of any pelvic cancer.

## Handling of discordant results

The accuracy of urogenital cytology will be measured against standard diagnostic tests for endometrial cancer. Concordance of cytology results between observers will be recorded, but primary analysis will consider 'ACUS', 'suspicious', 'adenocarcinoma' or 'malignant (others)' cytology results by consensus opinion to be a positive result. 'False negative' results, where standard tests identify endometrial cancer but urogenital cytology does not, will be reviewed to identify possible contributing factors (patient factors, tumour factors or test errors). Subject to ongoing consent, women will be resampled before they undergo hysterectomy, to determine whether repeat sampling is helpful for missed cases. This will be possible because cytology results will be reported prospectively, in 'real time' and on a weekly basis, wherever possible. Cytology review of missed cases will be carried out at a multiheaded microscope by the cytology team. Second±further slides will be prepared and reviewed where sufficient residual sample allows. 'False positive' cases, where urogenital cytology is positive but standard tests are not, will be handled carefully. If the diagnostic pathway (figure 2) has not been completed, the responsible clinician will be contacted and asked to consider further tests, for example, hysteroscopy or an MRI scan. If the malignant cells identified by cytology could have originated elsewhere, further tests may be warranted (eg, cystoscopy and colposcopy). Retrospective blinded review of 10% of cases, including test positive, test negative, complex and discrepant cases, will facilitate formal assessment of intraobserver and interobserver variability and whether or not there is evidence for a 'learning curve' effect.

## Assessment of adverse events

Adverse events arising during the study will be recorded and managed in accordance with standard clinical practice.

## Data management and monitoring

Data will be managed by a dedicated project manager to ensure validity, accuracy and reliability. Data will be handled in accordance with Good Clinical Practice and the Data Protection Act (2018). Participants will be assigned a study-specific ID. Written, informed consent will be obtained for all participants and forms stored in local site files within locked filing cabinets. Clinical data will be collected on case report forms (CRFs) and entries verified by inspection against source data. No patient identifiers will be stored in CRFs or on the study database. A sample of CRFs (10% or as per the study risk assessment) will be checked on a regular basis for verification of all entries made. Deidentified data will be stored on a study-specific REDCap database. The capture of data on the study database will be checked and verified. Where corrections are required, these will carry a full audit trail

and justification. The sponsor will periodically audit the study site file, a sample of CRFs, consent forms and source data and accuracy of the study database to ensure satisfactory completion.

## Statistical analyses

Statistical analyses will be carried out in R V.3.2.5 (R Development Core Team, Vienna, Austria) and overseen by the trial statistician. A STARD diagram depicting the flow of participants through the study will be presented. This will describe the number of participants who met the inclusion criteria but did not take part and reasons for this. Participants' demographical and clinical characteristics at baseline will be presented by final diagnosis using appropriate descriptive statistics, mean and SD for continuous measures that are approximately symmetrical median and quartiles if the distribution is skewed. Discrete outcomes will be described using both the number and proportion (percentage). The distribution of disease will also be presented for the index test by results of the reference standard.

Analysis of the primary objective will compare the clinical performance of cytology with the reference test. Thus, sensitivity, specificity, NPV, positive predictive value and diagnostic accuracy will be reported together with their corresponding 95% CIs, calculated using the exact binomial method. Where a histology result is not available, endometrial thickness <4 mm (histology not indicated) or discharge from diagnostic workup will be used as reference standard. Analysis of the secondary objectives will assess clinical performance of the urine test alone and the vaginal test alone for the detection of endometrial cancer in relation to the reference test. The clinical performance of urine and vaginal cytology for the detection of any cancer affecting the pelvic organs will also be calculated. Secondary analyses will follow the same approach as the primary analysis. Sensitivity analyses will consider the clinical performance of cytology with a broader definition of a positive cytology result (including glandular results as test positives). Interobserver agreement between the cytopathologists will be assessed using the kappa statistic and categorised as poor, fair, moderate and good. McNemar's $\chi^2$ test will be used to compare urogenital cytology with routine diagnostics. Multivariate logistic regression will evaluate the relationship between positive cytology and endometrial cancer diagnosis while adjusting for potentially predictive clinical characteristics like age and BMI. The diagnostic accuracy of urogenital cytology will be compared with individual elements of the standard diagnostic pathway, including TVS, hysteroscopy and endometrial biopsy. Consideration will be given as to where in the current diagnostic pathway for endometrial cancer urogenital cytology fits. Its use as a triage test prior to any other diagnostic workup both alone and in combination with clinical parameters (eg, age and BMI) will be modelled. We will also consider its usefulness in combination with transvaginal ultrasound scanning at various endometrial thickness cut-offs. The acceptability of

urogenital sampling will be compared with transvaginal ultrasound, hysteroscopy and endometrial biopsy (online supplemental appendix 1).

## Planned secondary use of the data and samples

Beyond the scope of this diagnostic accuracy study, we will use the prospectively collected data to assess the clinical performance of elements of the current diagnostic pathway, alone and in combination, including formal cost-effectiveness analyses, if resources permit. We will use the data to validate and compare the clinical performance of several published endometrial cancer risk prediction models and to develop a novel risk prediction model that includes urogenital cytology. We will report descriptive analyses of our cohort of women with PMB, including the distribution of risk factors and patterns of help-seeking behaviour. Residual urine and vaginal samples will be embedded in agar cell blocks or spun down and frozen for future translational research. Agar cell blocks will be used to identify cellular markers by immunohistochemistry that distinguish benign from malignant cells, for example, proliferation markers (Ki-67 and minichromosome maintenance (MCM) 2, which may facilitate their identification using adjunct immunocytochemistry, single-cell platforms or flow cytometry. Frozen cell pellets and supernatant fractions will be used to search for novel genomic, metabolomic and proteomic biomarkers.

## Ethics and dissemination

This study was adopted onto the National Institute for Health Research trial portfolio on 7 August 2018 and is sponsored by MFT. Any planned modifications to the protocol will be approved by the REC before they are adopted by the study. An audit trail of ethical amendments and documentation will be kept to allow monitoring by the research team and external regulatory bodies (table 2). The study was registered with an International Standard Randomised Controlled Trial Number on 9 August 2018.

## Study management

The study management group comprises the principal investigator, project manager, cytology team, clinical research fellows, research nurses and statistician, who will jointly monitor study conduct and progress. All aspects of the study and all study personnel will adhere to the study protocol (version 4.4 or subsequent approved version) and Good Clinical Practice and Data Protection principles. Regular team meetings will ensure quick resolution of recruitment issues, study processes and data collection inconsistencies.

## Patient and public involvement

The research question was developed in dialogue with patients, carers, members of the public and healthcare professionals. 'Which women with abnormal bleeding require specialist referral for investigation?' was ranked the second most important unanswered research question in the James Lind Alliance Womb Cancer Priority Setting Partnership, recognising the need for better diagnostic pathways for endometrial cancer.[19] 'What simple, non-invasive, painless, cost-effective and convenient tests can be used to detect cancer early?' was voted the most important research priority for detecting cancer early.[8] Urogenital cytology is simple, non-invasive and painless; whether it is effective for the detection of endometrial cancer is the focus of this study. We will disseminate our results through publication in peer-reviewed scientific journals; presentation at conferences; and via social media, blogs and charity websites.

## DISCUSSION

The DETECT Study will establish the diagnostic test accuracy of vaginal and urine samples for endometrial cancer detection by cytology in women with PMB. Urogenital cytology could offer a simple, acceptable, easy to administer test that could be used in primary care as a triage tool for women with unexplained PMB. Cytology positive women could be referred for diagnostic workup, while cytology negative women are quickly reassured without the need for unpleasant, invasive, anxiety-provoking tests, with massive cost-saving implications for the NHS.

**Author affiliations**
[1]Division of Cancer Sciences, The University of Manchester, Manchester, UK

| Table 2 | Summary of ethical amendments | |
|---|---|---|
| **Protocol** | **Date** | **Summary of changes** |
| V3 | 24.05.2018 | Funding and approval secured to expand the study to allow multicentre recruitment and NIHR portfolio adoption |
| V4.1 | 15.10.2018 | Additional exclusion criteria added:<br>1. Mirena coil in situ or removed within the last 3 months.<br>2. Any other condition that would compromise participant safety or data integrity. |
| V4.3 | 06.03.2020 | Approval to allow recruitment strategy to include proven endometrial cancer cases prior to hysterectomy |
| V4.4 | 16.07.2020 | Approval to allow remote consent and data collection, if required, as part of COVID-19 safety mitigation measures |

NIHR, National Institute for Health Research.

²Department of Obstetrics and Gynaecology, Manchester University NHS Foundation Trust, Manchester, UK
³Manchester Cytology Centre, Manchester University NHS Foundation Trust, Manchester, UK

**Contributors** The study was conceived and designed by EJC, who is the study guarantor. SC is responsible for the day-to-day running of the study, ethical/regulatory approvals and data management. ERJ, HOF, KN and CEB are responsible for patient recruitment and the collection of data and samples. NN, DS and DR are responsible for the cytological analyses. EJC provides study oversight and clinical guidance for the study. ERJ, SC and EJC drafted the manuscript. All authors contributed to, reviewed and approved the final version of the manuscript and contributed to the development and set-up of the study.

**Funding** This work is funded by the Jon Moulton Charity Trust. HOF is supported by a National Institute for Health Research (NIHR) Doctoral Research Fellowship (DRF-2018-11-ST2 054). KN is supported by a Cancer Research UK Manchester Cancer Research Centre Clinical Research Fellowship (C147/A25254) and the Wellcome Trust Manchester Translational Informatics Training Scheme. CEB is supported by a Manchester University NHS Foundation Trust (MFT) Clinical Research Fellowship. EJC is supported by the NIHR Manchester Biomedical Research Centre (IS-BRC-1215-20007) and an NIHR Advanced Fellowship (NIHR300650).

**Competing interests** None declared.

**Patient consent for publication** Not required.

**Ethics approval** This study was approved by the North West–Greater Manchester West Research Ethics Committee (REC reference: 16/NW/0660) and Health Research Authority (HRA) and Health and Care Research Wales (HCRW) on 5 September 2016. Subsequent amendments are detailed in table 2.

**Provenance and peer review** Not commissioned; externally peer reviewed.

**ORCID iD**
Emma J Crosbie http://orcid.org/0000-0003-0284-8630

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
