## [Reviewer comments · BMJ Open]

ARTICLE DETAILS

TITLE (PROVISIONAL)	DEveloping Tests for Endometrial Cancer deTection (DETECT): Protocol for a diagnostic accuracy study of urine and vaginal samples for the detection of endometrial cancer by cytology in women with postmenopausal bleeding
AUTHORS	Jones, Eleanor; Carter, Suzanne; O'Flynn, Helena; Njoku, Kelechi; Barr, Chloe; Narine, Nadira; Shelton, David; Rana, Durgesh; Crosbie, Emma

VERSION 1 – REVIEW

REVIEWER	Alcázar, Juan Clínica Universidad de Navarra
REVIEW RETURNED	12-Apr-2021

GENERAL COMMENTS	Interesting and well-designed study that addresses a relevant issue commonly found in clinical practice I have only minor comments 1. Why did you choose 5% prevalence? Endometrial cancer in PBM is a little bit higher 2. Do you consider exclude patients with previous history of cervical or ovarian cancer? 3. The 5th exclusion criterion is quite "open". Please be more specific 4. Which actions are planned to overcome problems with patients recruitment due to COVID-19 pandemic? 5. How many unsatisfactory results are expected? 6. Vaginal sampling is taken by the patient herself? If so, sampling quality could affect data obtained?
--

REVIEWER	Burbos, Nikolaos Norfolk and Norwich University Hospital
REVIEW RETURNED	15-Apr-2021

GENERAL COMMENTS	The authors present a well-written study protocol evaluating new diagnostic strategies for women with postmenopausal vaginal bleeding (PMB). The protocol is based on the results of a pilot study conducted by the same group (1). The study addresses an important clinical question and has the potential to improve the diagnostic pathways for women with PMB. The study population is clearly defined. The methods for the index and reference tests are described well. Patient and public involvement has also taken place. Statistical analysis is appropriate. The authors may wish to clarify the following points:
---

	1. Did the authors include patients with recurrent episodes of PMB during the study? If included, was each presentation with PMB considered as a new case? 2. Arranging investigations too soon after a negative initial assessment may result in duplication of some results. Have the authors considered the time interval to recommend further investigation, following a negative evaluation of women with PMB? 3. The authors plan to consider atypical endometrial hyperplasia as a positive result on sensitivity analysis. Can the authors explain the rationale for this? As the index test/tests aim to detect endometrial cancer, it may be more appropriate to consider atypical hyperplasia as a negative result. 1. O'Flynn H, Ryan NAJ, Narine N, Shelton D, Rana D, Crosbie EJ. Diagnostic accuracy of cytology for the detection of endometrial cancer in urine and vaginal samples. Nat Commun. 2021;12(1):952.
--	---

REVIEWER	Clarke, Megan National Cancer Institute
REVIEW RETURNED	16-Apr-2021

GENERAL COMMENTS	Endometrial cancer is the most common gynecologic malignancy diagnosed in the U.K., and like in many other countries, incidence rates of endometrial cancer in the U.K. have been on the rise. The majority of women diagnosed with endometrial cancer present with postmenopausal bleeding (PMB); however, PMB has many benign etiologies, and only 5-10% of women with PMB will be diagnosed with endometrial cancer. Current diagnostic approaches, including transvaginal ultrasound and endometrial biopsy, are costly and invasive. Moreover, 10-30% of endometrial biopsies yield insufficient or inadequate samples for diagnosis. The identification of minimally invasive, accurate testing approaches to rule out endometrial cancer among women with PMB would be of great clinical benefit. The protocol for the DEveloping Tests for Endometrial Cancer detection (DETECT) study aims to evaluate the diagnostic accuracy of urine and vaginal samples for endometrial cancer detection by cytology in women with PMB. The study will recruit approximately 2,000 women over a three-year period, with follow-up extending an additional year. I am highly supportive of this research but have some questions that if addressed would help clarify the protocol as written. Major points: 1. The greatest challenge I have with the protocol is understanding the clinical implications of the potential findings and how the investigators envision incorporating such a test into clinical practice. I realize that recommendations would be based on the performance of these tests, but I think it would still be good to consider some scenarios up front. a. For example, as described in the intro, this test is being proposed as a triage test for PMB that would safely reassure women testing negative. In the primary analysis, atypical hyperplasia is considered negative – does that mean women with PMB testing negative but with atypical hyperplasia would not undergo additional testing and be safely reassured? Would you recommend surveillance with repeat testing at a follow-up interval? 2. Hormone therapy use can lead to bleeding or spotting in postmenopausal women, particularly within the first 6 months of use. Do the investigators plan to account for PMB that might be
---

	secondary to HRT use, and therefore not necessarily associated with underlying malignancy? 3. The inclusion/exclusion criteria do not mention anything about previous endometrial biopsy or other prior diagnostic workup procedures. Do you anticipate having patients enrolled who may have been previously evaluated for PMB? Along those lines, will the investigators distinguish between women presenting with initial PMB versus recurrent given that recurrent PMB is associated with higher risk? 4. Will the investigators obtain data on stage at diagnosis? It would be interesting to know if both early and late stage tumors shed malignant cells into the lower genital tract at similar rates/amounts. Also, this will have important implications for determining whether this testing strategy could have clinical benefit in terms of early detection. 5. How will clinical follow-up of negative women be carried out? Will this be active or passive? For how long? Minor points:  1. Can the authors provide more detail regarding the rationale for the second urine collection? 2. Why aren't the investigators using the endometrial intraepithelial neoplasia diagnostic schema which is preferred to the WHO94? 3. Can the authors clarify the purpose of the multivariate model and what these associations would mean in terms of clinical practice? It seems like stratified analyses of predictive values by salient risk factors would be the preferred approach. 4. What is meant by "random endometrial biopsy" (page 12)? 5. It will be very important to calculate the complement of the negative predictive value corresponding to the risk of cancer in those testing negative. 6. The classification of endometrial cancers diagnosed within 3 months of discharge as false negative seems like a short time window. Do the authors think cancers diagnosed within 6-12 months e.g. would have not been prevalent at the time of evaluation?
--	---

VERSION 1 – AUTHOR RESPONSE

Reviewer: 1

Dr. Juan Alcázar, Clínica Universidad de Navarra

Comments to the Author:

Interesting and well-designed study that addresses a relevant issue commonly found in clinical practice

Thank you for your kind words.

I have only minor comments

1. Why did you choose 5% prevalence? Endometrial cancer in PBM is a little bit higher

We chose 5% based on an audit of postmenopausal bleeding at our centre.

2. Do you consider exclude patients with previous history of cervical or ovarian cancer?

The aim is to have a representative sample of women referred with unexplained postmenopausal bleeding. We will exclude women with a history of hysterectomy. Most women with a history of ovarian or cervical cancer will have had a hysterectomy. Those who have not are unlikely to be referred to a diagnostic service with 'unexplained postmenopausal bleeding', but rather undergo expedited follow up by their oncologist in the event of symptoms.

3. The 5th exclusion criterion is quite "open". Please be more specific

This is a requirement of our ethics committee and it is a 'catch-all' to ensure patient safety remains paramount throughout conduct of the study.

4. Which actions are planned to overcome problems with patients recruitment due to COVID-19 pandemic?

The COVID-19 pandemic caused temporary closure to recruitment. It also reduced the number of women referred to our centre with postmenopausal bleeding. When the study re-opened, the throughput of suitable patients fell, as did our capacity to recruit them due to staff sickness, staff pregnancy and the introduction of COVID-19 safety measures. To ensure we reached our target of 100 endometrial cancer patients within the study timeframe, we focused our recruitment effort on women most likely to have an underlying endometrial cancer diagnosis. For example, women with strong clinical risk factors (elderly, obese); women who had already had an ultrasound scan showing an endometrial thickness >4mm; and women with endometrial cancer awaiting hysterectomy.

5. How many unsatisfactory results are expected?

In the pilot study, we did not have many unsatisfactory results (<5%) and anticipate similarly small proportions in this larger study.

6. Vaginal sampling is taken by the patient herself? If so, sampling quality could affect data obtained?

In this study, vaginal sampling is carried out by a research practitioner immediately prior to standard diagnostics. This ensures sampling is carried out to a strict protocol and ensures reliability of the results. If the DETECT study shows clinical utility of this tool, the potential to expand its use to home-based vaginal self-sampling is an exciting prospect worthy of further study.

Reviewer: 2

Dr. Nikolaos Burbos, Norfolk and Norwich University Hospital

Comments to the Author:

The authors present a well-written study protocol evaluating new diagnostic strategies for women with postmenopausal vaginal bleeding (PMB). The protocol is based on the results of a pilot study conducted by the same group (1). The study addresses an important clinical question and has the potential to improve the diagnostic pathways for women with PMB. The study population is clearly defined. The methods for the index and reference tests are described well. Patient and public involvement has also taken place. Statistical analysis is appropriate.

Thank you for these generous comments.

The authors may wish to clarify the following points:

1. Did the authors include patients with recurrent episodes of PMB during the study? If included, was each presentation with PMB considered as a new case?

We will include a representative sample of women referred for investigation of postmenopausal bleeding in this study. This may include women with recurrent episodes of postmenopausal bleeding. However, each participant will only be recruited once to this study.

2. Arranging investigations too soon after a negative initial assessment may result in duplication of some results. Have the authors considered the time interval to recommend further investigation, following a negative evaluation of women with PMB?

We will only recruit women with recurrent symptoms once to the study. In practice, it is unusual for women to be re-referred as a new case of unexplained PMB <3 months after the last episode.

3. The authors plan to consider atypical endometrial hyperplasia as a positive result on sensitivity analysis. Can the authors explain the rationale for this? As the index test/tests aim to detect endometrial cancer, it may be more appropriate to consider atypical hyperplasia as a negative result.

Thank you. This is an important point. We do not consider atypical hyperplasia a positive result in this study and have modified the text to make this clearer.

1. O'Flynn H, Ryan NAJ, Narine N, Shelton D, Rana D, Crosbie EJ. Diagnostic accuracy of cytology for the detection of endometrial cancer in urine and vaginal samples. *Nat Commun.* 2021;12(1):952.

Reviewer: 3

Dr. Megan Clarke, National Cancer Institute

Comments to the Author:

Endometrial cancer is the most common gynecologic malignancy diagnosed in the U.K., and like in many other countries, incidence rates of endometrial cancer in the U.K. have been on the rise. The majority of women diagnosed with endometrial cancer present with postmenopausal bleeding (PMB); however, PMB has many benign etiologies, and only 5-10% of women with PMB will be diagnosed with endometrial cancer. Current diagnostic approaches, including transvaginal ultrasound and endometrial biopsy, are costly and invasive. Moreover, 10-30% of endometrial biopsies yield insufficient or inadequate samples for diagnosis. The identification of minimally invasive, accurate testing approaches to rule out endometrial cancer among women with PMB would be of great clinical benefit. The protocol for the DEveloping Tests for Endometrial Cancer detection (DETECT) study aims to evaluate the diagnostic accuracy of urine and vaginal samples for endometrial cancer detection by cytology in women with PMB. The study will recruit approximately 2,000 women over a three-year period, with follow-up extending an additional year. I am highly supportive of this research but have some questions that if addressed would help clarify the protocol as written.

Thank you for your support.

Major points:

1. The greatest challenge I have with the protocol is understanding the clinical implications of the potential findings and how the investigators envision incorporating such a test into clinical practice. I realize that recommendations would be based on the performance of these tests, but I think it would still be good to consider some scenarios up front.

a. For example, as described in the intro, this test is being proposed as a triage test for PMB that would safely reassure women testing negative. In the primary analysis, atypical hyperplasia is considered negative – does that mean women with PMB testing negative but with atypical hyperplasia would not undergo additional testing and be safely reassured? Would you recommend surveillance with repeat testing at a follow-up interval?

As you correctly point out, the positioning of this new test within the established diagnostic pathway needs to be defined. To some extent, this will depend on its clinical performance. A test with high sensitivity may work well as a 'rule out test' for women with PMB, ensuring urgent referral only of those at greatest risk of endometrial cancer, and safe reassurance of those at lowest risk. It would be important for the test to pick up biologically aggressive, type II tumours with high diagnostic accuracy in this scenario since these are the tumours with the poorest clinical outcomes. We do not expect atypical hyperplasia will be picked up by the test (it wasn't in the pilot study). We do not know enough about the biology of atypical hyperplasia to understand whether it is important to diagnose and treat them all, or whether this approach results in overtreatment of low risk lesions that would have regressed spontaneously with time. We would recommend repeat testing in the event of persistent symptoms. Repeat urogenital cytology plus a transvaginal ultrasound scan would enable women with persistent symptoms to be reassured for a second time, or triaged to hysteroscopy +/- biopsy. This should enable the diagnosis of all clinically relevant atypical hyperplasias (ie those that persist and/or progress) without picking up those lesions that are transient, do not cause repeated symptoms, and at low risk of malignant transformation.

2. Hormone therapy use can lead to bleeding or spotting in postmenopausal women, particularly within the first 6 months of use. Do the investigators plan to account for PMB that might be secondary to HRT use, and therefore not necessarily associated with underlying malignancy?

Thank you for raising this point. We think it's really important that the test can distinguish unscheduled bleeding on HRT from bleeding secondary to sinister underlying pathology. A high proportion of our referrals for urgent investigation of postmenopausal bleeding are HRT-related, and it is therefore important to include a representative sample of all comers, including those taking HRT, to ensure the tool is fit for purpose in its intended population.

3. The inclusion/exclusion criteria do not mention anything about previous endometrial biopsy or other prior diagnostic workup procedures. Do you anticipate having patients enrolled who may have been previously evaluated for PMB? Along those lines, will the investigators distinguish between women presenting with initial PMB versus recurrent given that recurrent PMB is associated with higher risk?

We will recruit all comers with postmenopausal bleeding in order to establish the utility of the tool in a standard referral population. This may include women who have previously been investigated for PMB, however, they will only be recruited once to this study. We will record if the participant has recurrent PMB, but we will treat the current episode of PMB as the index episode against which we will judge the accuracy of the tool.

4. Will the investigators obtain data on stage at diagnosis? It would be interesting to know if both early and late stage tumors shed malignant cells into the lower genital tract at similar rates/amounts. Also, this will have important implications for determining whether this testing strategy could have clinical benefit in terms of early detection.

Yes, absolutely, this is very important. We are collecting histological subtype, grade and stage of disease. We are interested in knowing that the tool not only picks up early stage disease but that it does not miss any advanced stage tumours either.

5. How will clinical follow-up of negative women be carried out? Will this be active or passive? For how long?

We do not have resources to actively follow up the test negative women. We will chase clinical outcomes of all women recruited to the study up until the point of discharge and monitor any subsequent re-referrals to our service.

Minor points:

1. Can the authors provide more detail regarding the rationale for the second urine collection?

In the pilot study, the first urine voided in the morning contained the most cancer cells, possibly due to pooling of shed tumour debris in the vagina overnight. Where possible, we want first void urine but in practice, women forget to collect it. The pilot study showed that immediately fixing urine with CytoRich Red gave the best results. Women bringing a sample of their first void urine to clinic may need to store it for several hours at room temperature and without preservative, which may affect its quality. A decision was therefore made to collect two urine samples, the first void of the day as the preferred sample, and a second sample on arrival at clinic, just in case the first sample is not suitable or forgotten. We did not want to miss the opportunity to have at least one urine sample per participant.

2. Why aren't the investigators using the endometrial intraepithelial neoplasia diagnostic schema which is preferred to the WHO94?

Our specialist gynaecological pathologists assess endometrial samples using hyperplasia with/without atypia as their preferred classification system. They do not use the EIN classification system.

3. Can the authors clarify the purpose of the multivariate model and what these associations would mean in terms of clinical practice? It seems like stratified analyses of predictive values by salient risk factors would be the preferred approach.

Yes, that's right. We envisage a scenario whereby a risk prediction model that incorporates clinical factors (eg age, BMI) and the results of certain diagnostic tests (eg endometrial thickness on transvaginal ultrasound, positive/negative cytology) could be used to triage women for reassurance or invasive diagnostic procedures.

4. What is meant by "random endometrial biopsy" (page 12)?

We mean a sample taken blindly from the uterine cavity using a pipelle endometrial sampler, as opposed to one taken under direct vision at hysteroscopy.

5. It will be very important to calculate the complement of the negative predictive value corresponding to the risk of cancer in those testing negative.

Yes, we agree. This will be a very important metric to consider, when assessing the value of urogenital cytology for endometrial cancer detection.

6. The classification of endometrial cancers diagnosed within 3 months of discharge as false negative seems like a short time window. Do the authors think cancers diagnosed within 6-12 months e.g. would have not been prevalent at the time of evaluation?

Essentially we are comparing urogenital cytology against the whole diagnostic pathway in routine clinical use. The current diagnostic pathway discharges women in whom tests fail to indicate endometrial cancer, and this is the reference standard against which we will establish a negative clinical outcome. We do not expect that atypical hyperplasia sheds malignant cells in the same way that endometrial cancer does (in the pilot study only 1 in 3 atypical hyperplasias had positive urogenital cytology). A woman with atypical hyperplasia [who tests negative for endometrial cancer by biopsy (reference standard) and cytology (index test)] may choose to undergo hysterectomy. However, this surgery may be scheduled routinely (rather than urgently) since endometrial cancer has not been diagnosed. A hysterectomy within three months of the index tests would likely show the pathology present at the time of these tests. But a hysterectomy many months later may show e.g. 'areas that just amount to G1 endometrioid endometrial cancer' that may not have been present at the time of the index tests. We have put the cut-off at 3 months to avoid calling these cases false negative, when we think it's equally likely that they have progressed from atypical hyperplasia to cancer in the interim.

VERSION 2 – REVIEW

REVIEWER	Burbos, Nikolaos Norfolk and Norwich University Hospital
REVIEW RETURNED	30-Jun-2021

GENERAL COMMENTS	Thank you for submitting a revised version of the manuscript and replying to the queries raised. I would like to point out that if the cohort is enriched with patients at high risk of endometrial cancer (ET>4 mm) or with patients diagnosed with endometrial cancer, this will affect the distribution of the predictors and cancer cases in the population. This will affect the validity of any predictive models developed. However, this will not be the case if the cohort is not 'enriched'.
---

REVIEWER	Clarke, Megan National Cancer Institute
REVIEW RETURNED	01-Jul-2021

GENERAL COMMENTS	The authors have adequately addressed the reviewers' comments. I wish them luck with this important study.
--